# Towards the Standardization of Photothermal Measurements of Iron Oxide Nanoparticles in Two Biological Windows

**DOI:** 10.3390/nano13030450

**Published:** 2023-01-22

**Authors:** Daniel Arranz, Rosa Weigand, Patricia de la Presa

**Affiliations:** 1Instituto de Magnetismo Aplicado (UCM-ADIF-CSIC), A6 km 22.500, 28230 Las Rozas, Spain; 2Departamento de Óptica, Facultad de Ciencias Físicas, Universidad Complutense de Madrid, Plaza de las Ciencias 1, 28040 Madrid, Spain; 3Departamento de Física de Materiales, Facultad de Ciencias Físicas, Universidad Complutense de Madrid, Plaza de las Ciencias 1, 28040 Madrid, Spain

**Keywords:** iron oxide nanoparticles, laser-induced photothermia, specific absorption rate

## Abstract

A systematic study on laser-induced heating carried out in two biological windows (800 nm and 1053 nm) for Fe_3_O_4_ nanoparticles in water suspension showed evidence of the strong dependence of the specific absorption rate (SAR) on extrinsic parameters such as the vessel volume or laser spot size. The results show that a minimum of 100 μL must be used in order to obtain vessel-size-independent SARs. In addition, at a constant intensity but different laser powers and spot size ratios, the SARs can differ by a three-fold factor, showing that the laser power and irradiated area strongly affect the heating curves for both wavelengths. The infrared molecular absorber IRA 980B was characterized under the same experimental conditions, and the results confirm the universality of the SARs’ dependence on these extrinsic parameters. Based on these results, we propose using solutions of IRA 980B as a standard probe for SAR measurements and employing the ratio SAR_iron oxide_/SAR_IRA 980B_ to compare different measurements performed in different laboratories. This measurement standardization allows us to extract more accurate information about the heating performance of different nanoparticles.

## 1. Introduction

Iron oxide nanoparticles have been largely investigated from a fundamental point of view [1] in regard to applications such as drug delivery [2], magnetic hyperthermia [3], and magnetic resonance imaging [4], among others. Recently, it was shown that iron oxide nanoparticles offer new possibilities regarding their optically activated heating capabilities for biomedical applications [5,6]. Whereas the heating efficiency of iron oxides under laser irradiation is much lower than that of the metallic ones, the former present several advantages: they can be synthesized with tailored structural, colloidal, and magnetic properties and, in addition, they are highly biocompatible and fully biodegradable within a few days [7]. Moreover, recent studies have shown that the heating efficiency of iron oxides under near-infrared irradiation can enhance the thermal stress supplied by an alternating magnetic field, which could benefit solid tumor removal [8,9,10,11,12].

The used iron oxides mainly encompass the ferrimagnetic magnetite (Fe_3_O_4_) and maghemite (γ-Fe_2_O_3_), both possessing good magnetic properties. The interest in photothermia mediated by iron oxide nanoparticles lies not only on the biocompatibility of the materials but also in the optical wavelength range in which they can be heated. On the one hand, the wavelength range covers the two biological windows, the first one ranging from 650 nm to 960 nm (with a peak of transmission at approximately 800 nm), in which skin, tissues, and hemoglobin present a minimum absorbance, and the second one ranging from 1000 nm to 1400 nm, which also corresponds to the water absorption bands [13,14,15]. On the other hand, near-infrared technology is already used in clinical practice [16,17].

Despite the abundant bibliography on this topic, the physical mechanism that optically activates the heating is still not clear, although it has been mainly attributed to the hot-phonon bottleneck effect [18] or the role of defect concentration [14].

Several studies have been performed in different laboratories with different particle sizes [12,14,19,20], shapes [12,14,20,21], clustering [12,13,14,20,22], coatings [6,8], iron oxidation states [13,20], irradiation intensities [11,18,20], and wavelengths [12,13,14,23], with the aim of determining the intrinsic and extrinsic parameters that maximize the heating efficiency under the infrared irradiation of the magnetic nanoparticles.

However, we are still missing a fundamental key that is required to understand the role played by the extrinsic and intrinsic physical parameters: the determination of a protocol that allows for the standardization of the measurements. For example, the measurements reported in the literature have been conducted using different colloidal containers (mainly Eppendorf tubes but also quartz cuvettes) [6,23], sample volumes (from 0.05 mL to 3.0 mL), and irradiation intensities (from 0.3 W/cm^2^ to 8.7 W/cm^2^) [6,13,14,18,23], without knowing the effects that these parameters have on the heating curves beyond the physical properties of the nanoparticles. It is worth noting that standardization protocols are a requirement for developing new techniques, especially those related to biomedical applications [24,25].

In this article, we show that the heating curves are affected not only by the properties of the nanoparticles but also by many parameters of the measurement procedure. For example, the volume is a key parameter, and we show that there exists a minimum volume of 100 µL that must be considered; otherwise, one can misinterpret the results. The laser intensity is also used to compare the different results, and we show here that samples irradiated at the same intensity but different laser powers and spot sizes lead to different heating efficiencies. In addition, the measurement of an Eppendorf tube or quartz cuvette surface with an infrared camera does not provide information about the heat loss of the colloid but rather the heat loss of the vessel surface.

From this information, it can be concluded that although a given author can provide values of the specific absorption rate (SAR) obtained in the experiments, it is not possible to compare these values with the results of other authors so as to conclude which kind of particle shows a more efficient heating under laser irradiation.

In this work, we use Fe_3_O_4_ nanoparticles to show that SAR values depend on the light power, light intensity, irradiated volume, vessel size, and beam size. Based on these results, we propose using an infrared absorber (IRA 980B) with absorption in two water windows as a reference in SAR measurements so that other authors can report their results in reference to it and, hence, a comparison between different iron oxide colloids in different experiments can be carried out.

## 2. Materials and Methods

Magnetite (Fe_3_O_4_) nanoparticles were synthesized by a modified co-precipitation method [26,27]. In the first step, a mixture of ethanol and distilled water in a 5:4 ratio were added to a three-necked, round-bottom flask. NaOH and KNO_3_ were added to obtain a basic solution at final concentrations of 70 mM and 100 mM, respectively. An acid solution was prepared separately, including 20 mL of 10 mM H_2_SO_4_ dissolved in water, and then FeSO_4_·7 H_2_O was added to obtain a final concentration of 24 mM. Once the mixtures in both the recipients were well-mixed, the acid solution was added rapidly to the basic solution under a N_2_ flow to prevent the oxidation of Fe_3_O_4_. The mixture turned dark green, which is commonly called green rust. The solution was vigorously magnetically stirred for 10 min. Thereafter, the flask was introduced into an oil bath at 90 °C for 24 h. The precipitated nanoparticles were collected in centrifugation tubes and washed with water three times at 9000 rpm in cycles of 15 min. The final colloid was black, showing a good stability. No precipitate of the nanoparticles was observed at the bottom of the container after 6 months stored in the refrigerator. The final concentration of the iron oxide Fe_3_O_4_ suspended in water was 20 mg/mL.

The sample was characterized by X-ray diffraction (XRD) using a Siemens D5000 diffractometer (from Siemens AG; Munich, Germany) with a Co Kα as the radiation source (λ = 1.78897 Å). The XRD pattern ranged from 10 to 110° with a 0.02° step. The diffraction patterns were analyzed using the PANalytical software X’Pert HighScore Plus (Version 2.2b), and the diffraction peaks were compared to the reference patterns. The crystallite size was calculated using the Scherrer formula (D = Kλ/βcosθ), where *K* is the form factor, λ is the radiation source’s wavelength, β is the full width at half maximum (FWHM), and θ is the Bragg angle.

The size and shape of the magnetic nanoparticles were characterized by transmission electron microscopy (TEM) using a JEOL JEM100 (100 kV) electron microscope (from Japan Electron Optics Laboratory Company Limited; Tokyo, Japan). The magnetic nanoparticles, in an aqueous solution, were dropcast onto a copper grid covered with a perforated carbon layer. The size distribution of the nanoparticles was obtained by measuring the diameters of more than 700 nanoparticles. The data were fitted to a log-normal curve, and the mean size and the standard deviation were obtained.

The hydrodynamic size and zeta potential of the suspended nanoparticles were measured by dynamic light scattering (DLS). This was performed using a Malvern Zetasizer, model Nano-ZS, equipped with an aqueous solution laser (λ = 632 nm), (from Malvern Panalytical; Malvern, United Kingdom).

The magnetic properties were studied employing a Quantum Design MPMS-5S SQUID magnetometer (from Quantum Design, Inc; California, United States of America). Hysteresis cycles of 50 kOe were performed at 5 and 300 K. The zero-field-cooled and field-cooled (ZFC-FC) curves were measured from 5 to 300 K at 100 Oe applied field.

IRA 980B, a commercial infrared absorber, was purchased from Exciton-Luxottica (from Exciton-Luxottica; Lockbourne, Ohio, United States of America). IRA 980B comes in powder form and is soluble in ethanol but not in water. To obtain a solvent similar to water, a solution was prepared close to its saturation point in ethanol at 16 mg/mL (the manufacturer specifies that its saturation point is 16.25 mg/mL at 25 °C). Then, 1950 μL of H_2_O was added to 50 μL of the ethanol solution. The stability of the solution was confirmed by absorption spectra obtained in different weeks. This mixture, therefore, had a percentage of 2.5% in regard to the volume of ethanol over the total volume of water. To perform the experiments described below, we used freshly prepared solutions in water, starting with the almost saturated one in ethanol.

In order to investigate the effects of the sample volume and vessel diameter on the SAR, Teflon vessels were fabricated with an external diameter of 1 inch and different inner diameters and heights. Teflon was chosen because it is easy to mechanize, has a low porosity, and does not absorb radiation at the two wavelengths studied here. The vessels had 5, 6, 7, and 8 mm-wide diameters and were 2, 3, and 4 mm in height (see Figure 1), providing twelve different volumes ranging from 39 to 201 μL, as can be seen in Table 1. After each use, the Teflon vessels were cleaned with aqua regia.

Since Eppendorf tubes are widely used in the field, we also used white transparent tubes with a volume capacity of 0.5 mL, filled with 150 μL of solution.

The absorption of the samples was analyzed in a wavelength range from 1100 nm to 400 nm with a 0.5 nm step, using a SHIMADZU UV-1603 spectrophotometer (from Shimazu Corporation; Kyoto, Japan). The samples were introduced into a fused silica cuvette with a 1 mm optical path at the same concentration as that used for the heating measurements.

Two different laser sources were employed to irradiate the samples, both operating in the biological windows. One of the lasers was a collimated horizontally polarized Ti:Sa oscillator operating in continuous-wave (cw) at 800 nm, and the other one was a collimated horizontally polarized Nd:YLF cw laser operating at 1053 nm. To irradiate the samples directly, without traversing any glass or plastic wall, which could lead to the misinterpretation of the results, two periscope arrangements were built (Figure 2a). The laser beam was steered with a first periscope formed of two 45° mirrors (M_1_ and M_2_) to a height of 15 cm above the optical table. Then, it was propagated to the mirror M_3_ at 45°, which sent the beam down to the optical table again. The Teflon vessels were positioned on an x-y mount for 1-inch optics to precisely irradiate the center. This was achieved by placing an iris with a 0.5 mm diameter and 1-inch external diameter inside the x-y mount (Figure 2) and a photodiode head below. The iris was moved in the x-y direction until the maximum power was reached (Figure 2b), meaning that the iris was positioned at the center of the laser beam. Then, the Teflon vessel with a 1-inch base diameter could be placed above the iris inside the 1-inch-wide x-y mount, and consequently, the laser beam could impinge exactly on the center of the vessel (Figure 2c). The Eppendorf tubes were positioned to be irradiated laterally to the wall or from the top (see Figure 2a).

To change the power of the beam, a linear polarizer (P) was inserted after the first periscope. The beam sizes were measured at the entrance of mirror M_3_ using the knife-edge procedure, and both lasers operated in the Gaussian mode TEM_00_. The beam size of the 800 nm laser was 1300 μm, while for the 1053 nm laser, it was 1900 μm. To change the spot size, telescopes were built using two lenses (L_1_, L_2_). In this way, a beam with a 1900 μm diameter was also available for the 800 nm laser, and a beam with a 1300 μm diameter was also available for the 1053 nm laser. From now on, we will refer to the 1300 μm diameter as size 1 (S1) and the 1900 μm diameter as size 2 (S2).

An infrared camera, Model E53 from FLIR (from Teledyne FLIR; Portland, Oregon, United States of America), was used to measure the temperature of the samples under laser irradiation. The camera was connected to a laptop, and the data were extracted using the software FLIR Tools+ (version 6.4.18039.1003). The camera temperature ranged from −25 °C to 125 °C, with a thermal resolution of 0.04 °C. In a typical heating measurement, 10 s are recorded before the laser radiation to obtain a baseline. Then, laser radiation is applied to the sample until it reaches an equilibrium temperature, which can be seen in real time using the camera software. Eventually, the laser beam becomes blocked, and the consequent cooling curve is registered. For the Teflon vessels, the camera was placed to record the temperature of the sample surface, while in the case of the Eppendorf tubes, the camera was placed laterally (as in most reported experiments) to record the temperature of the wall.

## 3. Results

### 3.1. Structural and Magnetic Characterization

Appendix A shows the spinel structure’s typical diffraction pattern, which corresponds to Fe_3_O_4_ (JCPDS 01-088-0866). The mean particle size was determined to be 27 ± 2 nm by means of the Scherrer formula.

Figure 3 shows a TEM image of the magnetic nanoparticles and the particle size distribution. The particle size distribution was fitted with a log-normal function. The mean particle size is 27 nm, with a low polydispersity degree (standard deviation/mean size) of 0.2. The sample’s low polydispersity and high crystallinity suggest a homogeneous particle size distribution, supported by the good agreement between the particle sizes derived by XRD and TEM.

The hydrodynamic size distribution of the nanoparticles is 113 ± 42 nm, which is a low aggregation size, considering the mean particle size of the particles (see Appendix A). In addition, the zeta potential is −29.2 ± 1.3 mV, providing the long-term stability of the dispersion.

The coercivity and saturation magnetization extracted from the hysteresis loops (Appendix A) are H_c_ = 507 Oe, M_s_ = 89 emu/g and H_c_ = 17 Oe, M_s_ = 84 emu/g at 5 and 300 K, respectively, indicating that the particles are at the limit of the superparamagnetism–ferromagnetism transition. The ZFC curve (Appendix A) shows the typical Verwey transition at 110 K, below the bulk temperature of 120 K [28]. The presence of the Verwey transition also confirms that the sample is Fe_3_O_4_. Additionally, the M_s_ of 84 emu/g at 300 K is characteristic of Fe_3_O_4_ of this size [29].

The Fe_3_O_4_ and IRA 980B absorption spectra, at the concentration used for the photothermal characterization, were measured using the spectrophotometer (Figure 4). The backgrounds of the water and cuvette (or 2.5% ethanol in the water and cuvette) were also measured and removed from the whole sample absorption. As observed, the absorbance of Fe_3_O_4_ is smaller than that of IRA 980B. It is worth noting that the absorbance A at 800 nm is smaller than at 1053 nm for both materials (A_Fe304_(800) = 0.3834; A_IRA 980B_(800) = 0.5721; A_Fe304_(1053) = 0.4632; A_IRA 980B_(1053) = 0.8794). Vertical black lines at 800 nm and 1053 nm (laser wavelengths) were also drawn.

### 3.2. Photothermal Results

#### 3.2.1. Fe_3_O_4_ Nanoparticles

1.Volume Dependence

Teflon vessels of different heights and diameters were used to analyze the influence of the volume on the heating rate under laser irradiation. As mentioned before, twelve different Teflon vessels were used, with a capacity that varied from 39 μL to 201 μL. In Table 1, the different volumes resulting from the height and inner radius of the Teflon vessels are summarized.

The measurements were performed in two biological windows with two lasers at 800 and 1053 nm on the twelve Teflon vessels. The laser spot sizes were adjusted by employing the corresponding lens system, if required, to obtain a size S2 (1900 μm). The laser power used was 50 mW, which resulted in an intensity of 1.76 W/cm^2^, a value close to those previously reported [12]. The aqueous colloidal Fe_3_O_4_ had a concentration of 1.4 mg/mL of Fe_3_O_4_ in water. The Teflon vessels were filled to their calculated volume using precision micropipettes, and the colloid was discarded at the end of each measurement. The measurements consisted of recording, with the infrared camera focused on the surface of the solution, 10 s before laser irradiation, 110 s with the laser impinging on the center of the vessel, and 30 s with the laser beam blocked by a screen. In this way, a baseline for the temperature reference was obtained and, hence, an adequate record of the heating up and cooling down curves was acquired. Once the temperature evolution was recorded, the SAR was obtained using the following equation (Equation (1)) [29]:(1)$$SAR=\frac{C_{p}\;\rho_{H_{2}O}}{\left[Fe_{3}O_{4}\right]}\cdot \frac{dT}{dt},$$
where *C_p_* (Jg^−1^ K^−1^) corresponds to the specific heat of the medium (in this case, water), $\rho_{H_{2}O}$ corresponds to the water density (g/L), and [Fe_3_O_4_] is the concentration (g/L) of the Fe_3_O_4_ aqueous colloid. The term *dT*/*dt* is the temperature increase per unit time. To obtain a reliable SAR, the temperature rate should be measured under adiabatic conditions, as has been well-established in the case of magnetic hyperthermia [30,31]. In the case of photothermia, it is not possible (or makes no sense) to develop an adiabatic system, and the measurements are performed under non-adiabatic conditions. This has the drawback that the SAR is underestimated [32], since the heat exchange with the medium is not taken into account. To avoid this, the experimental and analytical methods described below [33] can be used to quantify the losses from the non-adiabatic setup and include them in the calculation of the SAR.

The measured cooling curve contains the information about the thermal exchange with the surroundings. Extracting this contribution from the heating curve results in a graph, where it is possible to observe how the sample would behave in adiabatic conditions, in which the temperature rise has a linear dependence on time. An example of the procedure is shown in Appendix A.

Figure 5 shows the behavior of the SAR as a function of the volume. As can be seen, the SARs fluctuate at volumes smaller than 100 µL for both wavelengths, preventing consistent and comparable measurements at these small volumes. However, above 100 µL, the SARs become independent of the volume and stabilize at around 1850 W/g and 2150 W/g for 800 and 1053 nm, respectively.

The SAR is higher at 1053 nm, because the optical absorbance is also higher at this wavelength than it is at 800 nm, as shown in Figure 4.

In the literature, the most widely used container for photothermal characterizations is the Eppendorf tube. By irradiating (at 50 mW) 150 µL of the colloid in an Eppendorf tube, the SAR value is much smaller than the values obtained using Teflon vessels (see Appendix A). This result was obtained independent of whether the irradiation was performed from the top or through the wall of the Eppendorf tube and whether the analysis was performed at the top, center, or bottom of the hotspot observed by the camera on the tube wall. The SARs obtained were ${\mathrm{SAR}}_{{\mathrm{Fe}}_{3}O_{4}}^{\mathrm{Top}}$(800) = 190 W/g; ${\mathrm{SAR}}_{{\mathrm{Fe}}_{3}O_{4}}^{\mathrm{Side}}$(800) = 158 W/g; ${\mathrm{SAR}}_{{\mathrm{Fe}}_{3}O_{4}}^{\mathrm{Top}}$(1053) = 165 W/g; and ${\mathrm{SAR}}_{{\mathrm{Fe}}_{3}O_{4}}^{\mathrm{Side}}$(1053) = 142 W/g.

This already hints at the fact that the conditions of the measurements must be carefully chosen and described. Furthermore, not only could the type of vessel and total volume employed be decisive parameters, but the relative size of the laser beam, with respect to the area of the vessel in the plane of incidence, i.e., the intensity of the laser beam, could also play a role.

2.Study of the Effect of the Laser Intensity

Once the limit volume for consistent measurements has been determined, the laser intensity is another important factor to be discussed. The laser intensity (*I* in W/cm^2^) is calculated as the power deposited on an area, and this is commonly used as a reference for the measurement conditions [8,13,14] or as an extrinsic parameter for the evaluation of the SAR [10,20,23,34]. In order to determine whether *I* can be considered as an extrinsic parameter to compare different photothermia experiments, the laser power and the spot area were varied to obtain the same *I*. The main idea is to use two different areas, *A*1 and *A*2 (corresponding to spot sizes *S*1 and *S*2), and adapt the laser power in order to obtain the same *I*.

To observe the heating, five different intensities were selected, taking into account the maximum and minimum powers delivered by the lasers. Through the corresponding lens systems, the laser spot diameters were adjusted to 1900 μm and 1300 μm. Using these two different sizes allows one to acquire measurements using the same laser intensity, while providing almost a two-fold difference in power. A smaller area of incidence leads to a decrease in the power so as to maintain a constant intensity, and vice versa. The following Table 2 shows the powers and intensities used, which are within the range of values frequently used in the literature [10,13,18,22,23].

Figure 6 shows the SARs as a function of *I* for different power/area combinations at both the 800 nm and 1053 nm wavelengths in a Teflon vessel of 151 μL (8 mm inner diameter, 3 mm height). As can be seen, the SARs increase with the intensity for both wavelengths. The SARs for 1053 nm are higher than those for 800 nm, which can be explained by the fact that the absorbance at 1053 nm is higher (see Figure 4). They also increase with a higher irradiation power, *P*2 vs. *P*1. Both behaviors are reasonable, because the higher the intensity or the higher the power is, the larger the number of photons per unit time arriving at the sample will be. However, using the same intensity but different laser powers (and, hence, different spot areas) has a significant effect on the heating efficiency of the sample. These SAR differences between the *P*1/*A*1 and *P*2/*A*2 curves are more noticeable for 800 nm than for 1053 nm.

It is apparent that the curves *P*1/*A*1 and *P*2/*A*2 are almost parallel to each other. The slight difference could be due to the fact that when the beam spot is larger, the light is irradiating a larger area. Then, the proximity of the walls, possible convective effects, and different heat conduction effects can influence the measurements.

Section 1 and Section 2 demonstrate the large discrepancy between the results of SAR performed with the same kind of particles at the same concentration but with different values of the extrinsic parameters, such as the vessel volume, laser beam size, and laser power. This suggests that it is necessary to set a reference or a standard so that different authors can refer to this standard in their measurements, performing the measurements under the same experimental conditions as the nanoparticle colloids. In this way, they could interpret the results of their magnetic colloids with respect to the standard and compare the heating efficiency of the nanoparticles between different laboratories. This would allow them to identify the intrinsic photothermal properties of the nanoparticles.

For this purpose, we propose using the molecular infrared absorber IRA 980B, since it absorbs at 800 nm and 1053 nm (see Figure 4) and is commercially available.

#### 3.2.2. Infrared Absorber IRA 980B

In the previous sections, it was shown that numerous factors affect the results obtained in photothermia measurements. We therefore propose introducing a molecular compound as a standard probe (IRA 980B), as described and characterized in previous sections. This compound has the advantage that it is commercially available and, therefore, can standardize the characterizations. The objective, then, is to show that this absorber shows a similar behavior when heated with light in comparison to Fe_3_O_4_ and, hence, corroborate the roles of extrinsic parameters in SAR measurements and the need to use a reference.

1.Volume Dependence

As performed for the Fe_3_O_4_ nanoparticles in Section 3.2.1, the analysis of the volume effect was performed with IRA 980B. The twelve Teflon vessels where irradiated at the sample concentration described in Section 2, with 50 mW power and the spot diameter *S*2 for both the 800 nm and 1053 nm lasers.

Figure 7 shows the SAR values obtained as a function of the volume for both laser wavelengths. Again, SAR fluctuates at small volumes (smaller than 100 μL) but shows stabilization at higher volumes. This occurs at both wavelengths.

As can be noted, the difference between the SAR values at 1053 nm (16,200 W/g) and 800 nm (6200 W/g) are higher than they are in the case of Fe_3_O_4_. As mentioned for Fe_3_O_4_, this difference can be explained through the difference in the absorbance that the sample shows at different wavelengths. The differences in the absorbance of IRA 980B are higher than those of Fe_3_O_4_, which can lead to larger differences in the SAR.

The fact that this stabilization effect occurs for both samples (IRA 980B and Fe_3_O_4_) suggests that the volume of the sample is a critical factor to be considered during the measurement planning. Volumes lower than 100 μL cause the SAR to be strongly dependent on the recipient shape, rendering the comparison with other compounds impossible.

As for Fe_3_O_4_, the heating curves were also recorded, irradiating the Eppendorf tubes from the top and from the side (Appendix A) with 50 mW power using 150 μL of the IRA 980B solution. Again, the SARs obtained are different from those obtained using the Teflon vessels and also differ depending on whether the irradiation is applied from the top or through the tube wall (see Appendix A). The SARs obtained are ${\mathrm{SAR}}_{\mathrm{IRA}\;908B}^{\mathrm{Top}}$(800) = 1194 W/g; ${\mathrm{SAR}}_{\mathrm{IRA}\;908B\;}^{\mathrm{Side}}$(800) = 750 W/g; ${\mathrm{SAR}}_{\mathrm{IRA}\;908B}^{\mathrm{Top}}$(1053) = 727 W/g; and ${\mathrm{SAR}}_{\mathrm{IRA}\;908B}^{\mathrm{Side}}$(1053) = 452 W/g. These results, again, support the fact that parameters that are extrinsic to a material system and can be optically activated have strong influences on the heating curves and, consequently, on the SAR values.

2.Study of the Effect of the Laser Intensity

To verify whether the intensity, power, and irradiated area also play a role in heating a molecular solution, the same experiment as that described in Section 3.2.1 was carried out using the power, areas, and intensities in Table 2. The results are shown in Figure 8 and, again, the higher the intensity is, the higher the SAR will be at both wavelengths. As can be seen, for a given intensity, the SAR is again higher for a higher power and spot area.

In view of the results obtained, we propose using IRA 980B as the standard probe for the photothermal characterization of iron oxide nanoparticles. The iron oxide nanoparticles and IRA 980B should be measured in the same experimental conditions, and the ratio SAR_iron oxide_/SAR_IRA 980B_ should be calculated in the way shown in Figure 9. As can be seen, this ratio is essentially independent of the intensity and indicates that the particles are 15% efficient with respect to the IRA 980B using laser irradiation at 800 nm, and the efficiency is of the order of 25% using 1053 nm. This allows us to identify the efficiency of the particles independent of the conditions of the power, spot area, and intensity of the laser.

This is not surprising, since an analysis of the SAR divided by the intensity, SAR/*I* (see Appendix A), also shows constant values as a function of the intensity, revealing the independence of the SAR from the set of parameters including the power, area, and intensity of the laser. We can conclude that this method of reporting results is much more informative than reporting only the SAR values of the nanoparticles.

## 4. Discussion

The results show that there are several critical parameters that can lead to heating curves which do not represent the physical properties of the Fe_3_O_4_ nanoparticles, but these are influenced by secondary effects due to the measurement procedures.

For example, the dimensions of the vessel are crucial for the proper determination of SAR. If the diameter or the height of the vessel results in a volume below the critical size, it can induce SAR values that strongly depend on the dimensions of the vessel, preventing one from obtaining information about the nanoparticle physical properties involved in the photothermal effect. This effect is probably due to border effects or heat diffusion and the interactions with the vessel walls.

Another relevant parameter is the laser intensity, determined as the laser power divided by laser spot area (*I = P/A*), which is normally given as an extrinsic parameter of the measurement [11,18,20]. If the intensity is truly an extrinsic parameter, the SAR should be the same for *P*1*/A*1 and *P*2*/A*2, because the laser intensity is the same. However, we showed that the higher the laser power and spot diameter are, the higher the SAR will be (see Figure 6). This could be an effect due to the ratio between the laser spot size and the vessel diameter. It is worth noting that the SAR values vs. intensity obtained with the 1900 µm beam size are quite similar at 800 nm and 1053 nm, whereas when the laser spot is smaller (1300 μm), the SAR is almost two-fold higher at 1053 nm than 800 nm. Since the absorbance is higher at 1053 nm than 800 nm, it is expected that the SAR increases considerably for the first wavelength. These experiments confirm that the size of the laser spot plays a relevant role.

As noted previously, most of the photothermia experiments of iron oxide nanoparticles are performed either in Eppendorf tubes [11,13] or quartz cuvettes [6,23]. In this work, we compared the SARs for the colloids in Eppendorf tubes with those for the containers designed for this work. The Eppendorf tube was filled with 150 μm of the nanoparticle colloid at the same concentration as that used for the Teflon vessels. The Eppendorf tube was irradiated from the top, as most experiments show [9,13], and from the side. As can be seen in Appendix A, the heating rates of the colloid are much smaller than those for the vessels designed for these experiments. It is worth noting that the SARs at 1053 nm are smaller than those at 800 nm for both the top and side irradiation, contrary to what is observed in the Teflon vessels. These results suggest that the material of the Eppendorf tube affects the heating dissipation of the irradiated Fe_3_O_4_ nanoparticles.

Considering all these results, it is evident that there is a need for a standardization sample to obtain a better understanding of the physical properties of the iron oxide nanoparticles under laser irradiation. To this end, we propose the solution of IRA 980B, a commercially available molecular infrared absorber. This material was also investigated as a function of the volume, laser spot area and power, and wavelength. This sample has a volume dependence similar to those of the Fe_3_O_4_ nanoparticles, i.e., the SAR remains almost constant for a volume larger than 100 μL and fluctuates below it. Regarding the dependence on the intensity, it also depends on the laser power and spot area, as can be seen in Figure 8, showing that this effect is related not only to the Fe_3_O_4_ but also to other materials. On the other hand, there are differences in the SAR values, since the differences in the absorbance at 800 and 1053 nm are much higher in the case of IRA 980B than in Fe_3_O_4_. Therefore, the SAR values differ considerably for each wavelength.

Regarding the characterization in the Eppendorf tubes, IRA 980B shows a similar behavior when compared to the results obtained for Fe_3_O_4_. The SAR at 1053 nm is smaller than the SAR at 800 nm, which supports the idea that the Eppendorf tube material affects the absorbance of the laser irradiation at different wavelengths.

Reporting the results of the ratio SAR_iron oxide_/SAR_IRA 980B_ is highly informative, since it allows us to obtain a standard with which to compare results obtained in different laboratories for different particles and measurement conditions.

## 5. Conclusions

The heating efficiency of an Fe_3_O_4_ colloid under laser irradiation was investigated as a function of the vessel volume, laser power, and spot size. We showed that these parameters can affect the photothermal curves and do not necessarily provide information on the physical properties of the nanoparticles.

The samples were irradiated at 800 nm and 1053 nm within two biological windows. The results showed that, for both wavelengths, the SARs are independent of the vessel size at volumes higher than 100 μL. The size of the laser spot is also relevant, since, for large spot sizes, secondary effects can affect the heating curves, i.e., at the same laser intensity, the SAR depends on the laser spot size. In addition, we observed that the measurements performed in an Eppendorf tube also affect the heating curves and, consequently, the SAR values.

We provided evidence showing that the geometrical parameters affect the photothermal characterization of the Fe_3_O_4_ colloid, which prevents the comparison between the results of different authors and an understanding of the physical properties involved in this phenomenon. Thus, we considered the need to introduce a molecular system that is able to release heat in the two biological windows so that authors compare their results for iron oxide nanoparticles with a standard reference under the same experimental conditions. An interesting proposal is the commercial IRA 980B as a reference probe. We recommend reporting SAR results as the ratio of the SAR of the nanoparticle divided by the SAR of IRA 980B. We also recommend recording the temperature rise, focusing directly on the irradiated surface, and providing information about the colloidal volume, laser power, and spot size to obtain complete information about the measurement conditions.

## Figures and Tables

**Figure 1 nanomaterials-13-00450-f001:**
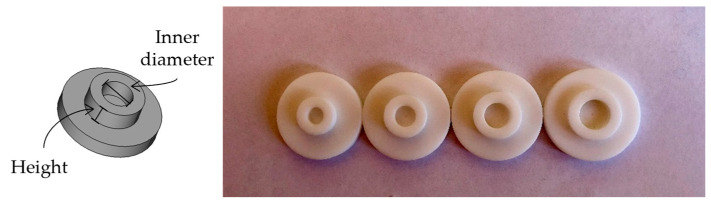
The 3D design and construction of the Teflon vessels with different inner diameters and heights.

**Figure 2 nanomaterials-13-00450-f002:**
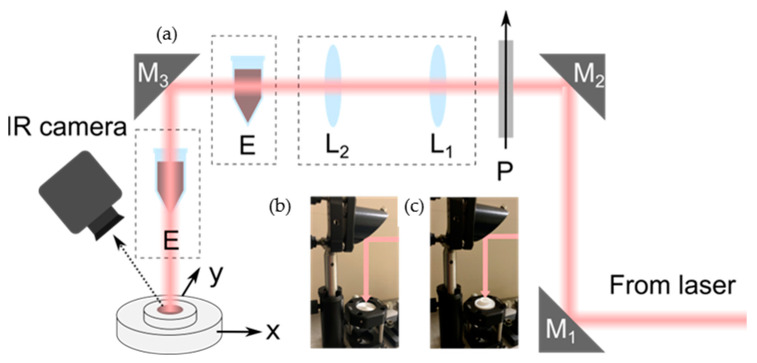
(**a**) Optical set-up for the heating measurements. (**b**) The x-y mount with the alignment iris. (**c**) The x-y mount with the Teflon vessel on top. P: Polarizer; L1, L2: Lenses; E: Eppendorf tubes. The boxes in dashed lines indicate that the elements are set in the light path when required.

**Figure 3 nanomaterials-13-00450-f003:**
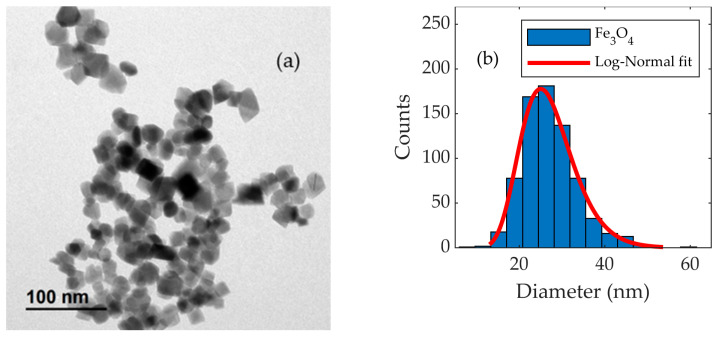
(**a**) Typical TEM image of the Fe_3_O_4_ nanoparticles and (**b**) TEM image analysis resulting in the histogram, fitted with a normal distribution.

**Figure 4 nanomaterials-13-00450-f004:**
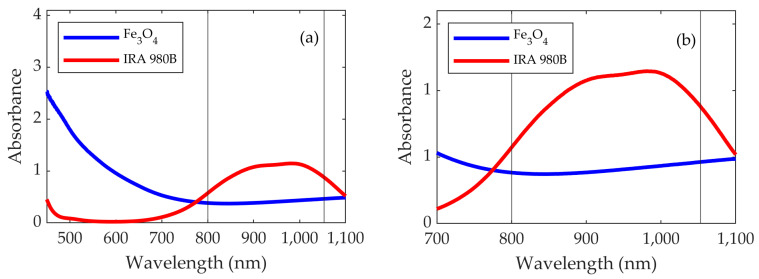
(**a**) Absorbance of the Fe_3_O_4_ (blue) and IRA 980B (red). (**b**) Zoom on the near-infrared area, where the lasers operate. Vertical black lines correspond to the laser wavelengths at 800 nm and 1053 nm.

**Figure 5 nanomaterials-13-00450-f005:**
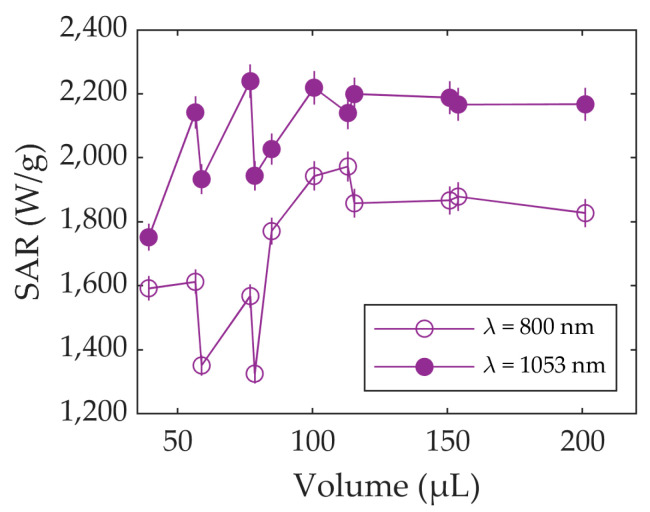
SAR vs. volume for Fe_3_O_4_ using lasers at 800 nm and 1053 nm. Errors are represented as vertical bars.

**Figure 6 nanomaterials-13-00450-f006:**
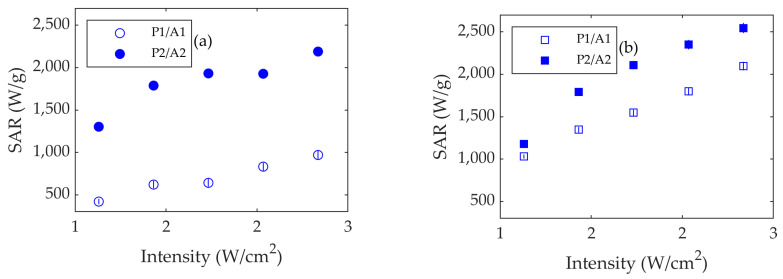
(**a**) SAR vs. intensity for Fe_3_O_4_ using lasers at 800 nm and (**b**) 1053 nm. Open dots and squares stand for laser diameter S1 (1300 μm), and full dots and squares stand for S2 (1900 μm). *A*1 and *A*2 are the laser spot areas corresponding to *S*1 and *S*2. Errors are represented as vertical bars.

**Figure 7 nanomaterials-13-00450-f007:**
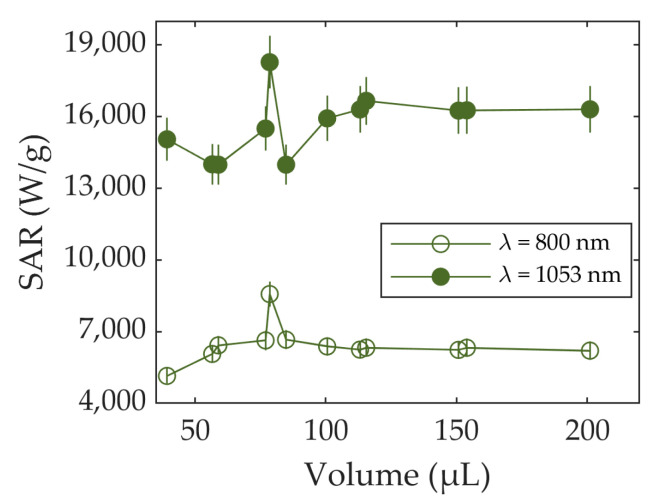
SAR vs. volume for IRA 980B using lasers at 800 nm and 1053 nm. Errors are represented as vertical bars.

**Figure 8 nanomaterials-13-00450-f008:**
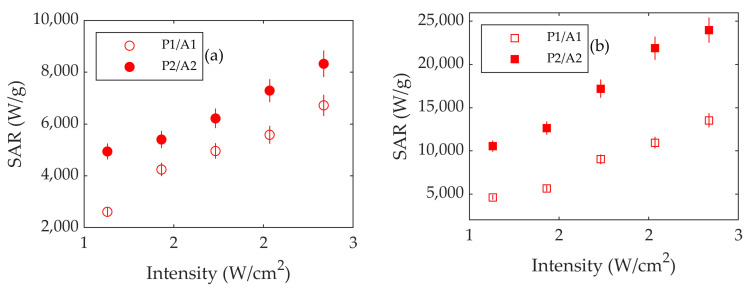
SAR vs. intensity for IRA 980B using lasers at (**a**) 800 nm and (**b**) 1053 nm. Open dots and squares stand for laser diameter *S*1 (1300 μm), and full dots and squares stand for *S*2 (1900 μm). *A*1 and *A*2 are the laser spot areas corresponding to *S*1 and *S*2. Errors are represented as vertical bars.

**Figure 9 nanomaterials-13-00450-f009:**
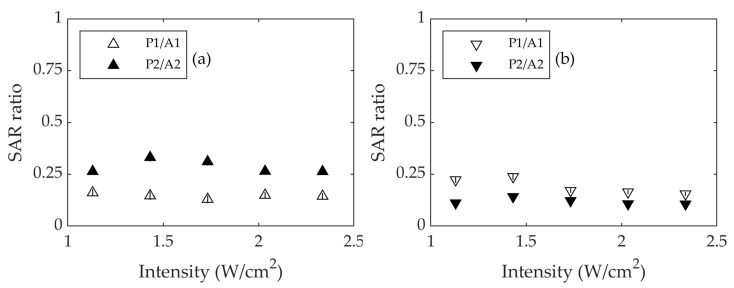
Ratio SAR_iron oxide_/SAR_IRA 980B_ using lasers at (**a**) 800 nm and (**b**) 1053 nm. Open triangles stand for *S*1 (1300 μm), and solid triangles stand for *S*2 (1900 μm). *A*1 and *A*2 are the laser spot areas corresponding to *S*1 and *S*2. Errors are represented as vertical bars.

**Table 1 nanomaterials-13-00450-t001:** Volume of the different Teflon vessels, deduced from the heights and inner radius.

Volume (μL)	Height (mm)		
Inner Diameter (mm)	2	3	4
5	39	59	79
6	57	85	113
7	77	115	154
8	101	151	201

**Table 2 nanomaterials-13-00450-t002:** Powers used according to the spot diameter (area) to obtain a given intensity of laser radiation.

Spot Diameter (μm); Spot Area (cm^2^)	*S*1 = 1300; *A*1 = 0.0133	*S*2 = 1900; *A*2 = 0.0284	
	*P*1 (mW)	*P*2 (mW)	*I* (W/cm^2^)
	15	32	1.13
	19	41	1.43
	23	49	1.73
	27	58	2.03
	31	66	2.34

## Data Availability

Not applicable.

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
