# Peer review of "Towards the Standardization of Photothermal Measurements of Iron Oxide Nanoparticles in Two Biological Windows"

_nanomaterials, 2023, doi:10.3390/nano13030450_

Round 1

Reviewer 1 Report

This manuscript reports the systematic study on laser-induced heating carried out in two biological windows (800 nm and 1053 nm) for 27 nm Fe3O4 nanoparticles in water suspension. The solutions of IRA 980B is suggested as a standard probe for SAR measurements and to employ the ratio SARiron oxide/SARIRA 980B to compare different measurements performed in different laboratories. SARs are independent of the vessel  size for volumes higher than 100 μL. The size of the laser spot is also relevant since, for large spot sizes, secondary effects can affect the heating curves, i.e, for same laser intensity the SAR depends on the laser spot size. The results are very good and interesting. However, some points of the manuscript should be improved. Specific comments are given below.

1.    The Fe3O4 nanoparticles should be measured by XPS.

2.    The hydrodynamic size and zeta potential of Fe3O4 nanoparticles should be measured.

3.    The photothermal conversion efficiency of Fe3O4 nanoparticles should be measured.

4.    The results of Fe3O4 nanoparticles should be compared with other reports.

5.    Please carefully check the text for writing and grammar.

Reviewer 2 Report

In this work, the systematic study on laser-induced heating carried out in two biological windows (800 nm and 1053 nm) for Fe3O4 nanoparticles in water suspension shows evidence of the strong dependence the specific absorption rate (SAR) has on extrinsic parameters such as vessel volume or laser spot size. Their work showed the evidence that there exist geometrical parameters that affect the photothermal characterization of Fe3O4 colloids that prevail the comparison of results between different authors and the understanding of the physical properties involved in this phenomenon. An interesting proposal is the commercial IRA 980B as a reference probe for further SAR measurements/applications. This measurement standardization will also allow to extract more accurate information about the heating performance of different nanoparticles.

1.       Title: Delete “.”

2.       Abstract: If we check the TEM images and corresponding particle analysis, the distribution of Fe3O4 nanoparticles is not monodisperse (the particles showed a polydispersity degree). Thus, I think the ~27 nm diameter should not be mentioned, or should add the average number

3.       Will the photothermal results/trend change if the concentration of Fe3O4 colloids increase, compared with 1.4 mg/mL?

4.       Figure 5-9. If all of these tests have repeated several times, the error bars should be added.

5.       In the Materials and Methods, what is Fe2SO4·7H2O? Did authors mean FeSO4·7H2O? Please check the synthesis steps carefully.

6.       Keywords should be checked again. Fe3O4 nanoparticles and Iron oxide nanoparticles, SAR and Specific absorption rate are redundant. Authors can delete one of them or add new keywords in the main article.

7.       What if we change Teflon to plastics or elastomers? If Teflon vessels are required to this test, the authors should highlight it or give a note in the experimental section.

Round 2

Reviewer 2 Report

All of my raised comments were well addressed, thus the acceptance is recommended.